complexity/behaviour/theoretical biology

game theory, behavioural experiments, imitation, network reciprocity

**Author for correspondence:**
Jelena Grujić
email: jelenagr@gmail.com

# Do people imitate when making decisions? Evidence from a spatial Prisoner's Dilemma experiment

## Jelena Grujić[1,2] and Tom Lenaerts[1,2]

[1]AI Laboratory, Vrije Universiteit Brussel, Brussels, Belgium
[2]MLG, Université Libre de Bruxelles, Brussels, Belgium

(iD) JG, 0000-0002-2688-6191; TL, 0000-0003-3645-1455

How do people decide which action to take? This question is best answered using Game Theory, which has proposed a series of decision-making mechanisms that people potentially use. In network simulations, wherein games are repeated and pay-off differences can be observed, those mechanisms often rely on imitation of successful behaviour. Surprisingly, little to no evidence has been provided about whether people actually imitate more successful opponents when altering their actions in that context. By comparing two experimental treatments wherein participants play the iterated Prisoner's Dilemma game in a lattice, we aim to answer whether more successful actions are imitated. While in the first treatment, participants have the possibility to use pay-off differences in making their decision, the second treatment hinders such imitation as no information about the gains is provided. If imitation of the more successful plays a role then there should be a difference in how players switch from cooperation to defection between both treatments. Although, cooperation and pay-off levels do not appear to be significantly different between both treatments, detailed analysis shows that there are behavioural differences: when confronted with a more successful co-player, the focal player will imitate her behaviour as the switching is related to the experienced pay-off inequality.

# 1. Introduction

Since the seminal work of Nowak & May [1,2], spatial structure has been proposed as a prominent mechanism to explain the presence of cooperation in social dilemmas like the Prisoner's Dilemma (PD). This early spatial research has been integrated into the narrative of how networks influence the level of cooperation [3–9], gaining traction as the network reciprocity mechanism [10,11]. Within these theoretical models, the assumption is made that players alter their

behaviour using a form of imitation, i.e. players copy (conditionally, unconditionally, probabilistically, …) the behaviour of more successful neighbours [12–15]. This *imitation* mechanism typically leads to more cooperation in networks when clusters of cooperators are formed, a dynamic referred to as *assortment* [16]. Notwithstanding the interesting properties of these results, they crucially depend on the assumption that imitating successful partners is the core mechanism to update players behaviour.

As shown in, for instance, Sysi-Aho *et al.* [17] and Van Segbroeck *et al.* [18], using different rules for the evolution of strategies may lead to very different outcomes. Moreover, the previously listed theoretical results appeared to be difficult to reproduce experimentally: behavioural experiments performed to validate if network structures promote cooperation, report levels of cooperation lower (around 20–30%) than observed in simulations [19–23]. Nonetheless these levels appear to be higher than what can be expected from elementary forms of stimulus-response learning [18]. These results put into question whether imitation is relevant for explaining changes in actions in networked games: in some experiments no evidence for imitation was found [21] while in others the evidence was borderline significant [24]. One experiment even concluded that reputation drives cooperation in network experiments, not imitation [25].

Experiments specifically designed to test the presence or absence of imitation in networked societies barely exist. The experimental work by Kirchkamp & Nagel is an exception [26]. They compared two treatments, one where players have information about the pay-offs of each of their neighbours and the other where they know only the average pay-off of cooperators and defectors in their neighbourhood. They found no difference between the two treatments. Also, they observed lower levels of cooperation than theoretically expected. This negative result may have been owing to the fact that in both treatments pay-off information was provided (exact versus average pay-offs), meaning that participants could use in both cases information related to pay-off differences to update their behaviour.

In economic literature, the theoretical basis for learning dynamics was developed for economic markets [27]. Vega-Redondo [28] provided the first theoretical work on imitation in this context, using imitation of the best player among the competitors as an update rule, a rule that can be considered equivalent to the 'imitate the best' rule by Nowak & May [1]. Later, Schlag [29] made his analysis using a 'proportional imitation rule', analogous to [14]. In contrast with the work of Vega-Redondo, players are not trying to learn from their competitors (or neighbours in a spatial game theory context), but from the players who have the same role in other groups. Just as the alternative update rules mentioned earlier, these rules lead to very different results. In addition, the experimental evidence for imitation appears to be inconclusive also: some experiments find confirmation for imitation (e.g. [30]) but others refute the possibility (e.g. [31]).

The seminal work of Apesteguia *et al.* [32] nonetheless provides some essential answers. They compared the outcomes of three experimental treatments of a linear Cournot market game, albeit not in a spatial configuration. Their experiments built on top of the earlier work of Schlag to examine the influence of who is being imitated. The authors showed that imitation occurs when knowledge about the success of competitors as opposed to non-competitors is used and that it depends on the difference between one's own pay-off and that obtained by these competitors.

Notwithstanding these insights, it is not clear whether these conclusion also hold for networked social dilemmas like the spatial PD. More importantly, in Apesteguia *et al.* [32] as well as in Kirchkamp & Nagel [26] participants always receive pay-off information, either about competitors or non-competitors, which raises the question of whether their observations depend on the source or the lack of specific pieces of information. Recent experimental work appears to indicate that their conclusions may not hold. Even worse, these experiments appear to suggest that the success of the neighbours (and as a consequence imitation) may not be relevant at all for the decision process. Moody conditional cooperation [21,23], which defines behavioural change in function of the number of cooperating neighbours and the previously taken action, was proposed as the key mechanism for human participants to change their actions from cooperation to defection and vice versa. Given these conflicting conclusions, it is crucial to provide a definitive answer through a new experiment within the context of spatial social dilemmas.

We, therefore, present here an analysis of the results of two experimental PD treatments on small lattices. They aim to verify whether there is a difference in the updating of the participant behaviour when they have access to both the neighbours' actions and the amounts they gained (treatment with information (TWI)), and when they only receive information about the neighbours' actions (treatment without information (TWO)). Different from prior work, we focus here on the presence or absence of information and not who is providing the information. Whereas in the first treatment players can decide to imitate more successful opponents or act in some other way according to this additional piece of information, they cannot in the second. We show, unlike [26], that the presence of pay-off information leads participants in a spatial PD to imitate their more successful neighbours and that, as in [32], this effect is stronger the bigger the pay-off difference.

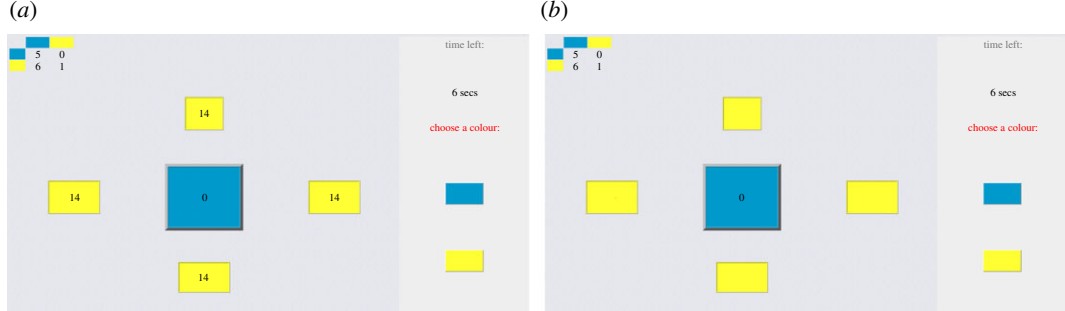

**Figure 1.** The game screens each player sees during the game. (*a*) Treatment with information (TWI). (*b*) Treatment without information (TWO). We see that the only difference is that in TWI they know how much their neighbour earned, i.e. the value in the yellow squares.

## 2. Methods

The experiment consisted of two treatments wherein an iterated PD was played on a square lattice with von Neuman neighbourhood (four direct neighbours). The lattice was of size $4 \times 4$ and had periodic boundary conditions, meaning that each session within a treatment consisted of 16 participants. Each spatial PD session was run for 50 rounds in order to acquire sufficient observations to produce good statistics and to provide sufficient time for cooperation to take off.

To avoid framing the participants' decision process with words like cooperation and defection, participants would choose between two colours: yellow (defection) and blue (cooperation). The rewards they could acquire in every round of the game were explained in function of the combination of these colours, as visualized by the following matrix:

|        | blue | yellow |
| ------ | ---- | ------ |
| blue   | 5    | 0      |
| yellow | 6    | 1      |

The pay-off values were selected to promote cooperation. In Rand *et al.* [33], it was postulated that this can be done in networks on the condition that the benefit-to-cost ratio is larger than the node degree in the network, an observation that was confirmed by Li *et al.* [34]. The benefit-to-cost ratio was defined in terms of the PD pay-off parameters, i.e. $T$, $R$, $P$ and $S$, as follows: $Q^* = (P + S - R - T)/(R + S - P - T)$. The selected pay-off matrix, shown above, produces a $Q^* = 5$, which is bigger than the node degree (i.e. $k = 4$) in the current experiment. Based on this condition a level of cooperation between 45 and 60% was expected to be observed [33].

Note that this condition is actually not met in the majority of the experiments mentioned in the introduction, potentially explaining why cooperation was only observed at highly reduced levels. However, Gracia-Lázaro's experiment [22] meets this requirement and still revealed low levels of cooperation, which puts into question Rand *et al.*'s postulate. We, therefore, also reanalysed our data using the methods of [33] to verify whether their conclusions could be confirmed in the context of the current experiment (see Results section).

In each round of the game, the colour selected by a participant was used to play against each of the four neighbours independently. Figure 1 shows the screen that each participant saw during the experiment: in the top left corner, the pay-off table for each colour combination is repeated. In the main part of the screen, five rectangles are shown, where the central one is the participant and the four squares around it represent the neighbouring players. The layout of the screen does not change during the experiment. The right column contains two buttons and the time remaining to make a choice (i.e. 30 s). Prior observations on the distribution of reaction times showed that this distribution has an exponentially decreasing tail [35] and that participants take an order of magnitude smaller than 30 s to make their choice, which was also observed in the current experiment (see the electronic supplementary material). It is also important to note that when the time ran out nothing happened, except that the participant was made aware that she should quickly make her choice (i.e. she received a warning on the screen). Participants were clearly informed about this before the start of the session.

**Table 1.** Generalized linear model of the cooperation level within both treatments (glm( formula = action $\sim$ round + info+ rnd + example, family = binomial(`logit'))). (The analysis shows that both round and info are significantly associated with the predicted cooperation level. The variables rnd and example have no significant correlation, as is required.)

| variable | estimate | std. error | z-value | Pr(|z|) |
|---|---|---|---|---|
| (intercept) | −0.189457 | 0.065447 | −2.895 | 0.00379** |
| round | −0.007853 | 0.001807 | −4.347 | $1.38 \times 10^{-05}$*** |
| info | −0.325525 | 0.052107 | −6.247 | $4.18 \times 10^{-10}$*** |
| rnd | 0.052922 | 0.052071 | 1.016 | 0.30947 |
| example | −0.063569 | 0.060368 | −1.053 | 0.29233 |

We only asked them to be prompt as delays in choosing would hinder the progress. To avoid end-of-game effects, we informed each participant that they would play multiple rounds, but not how many. Additionally, they were never informed in what round they were, so it was difficult for them to keep track of the rounds.

Figure 1 shows the difference in setup between the two treatments. In the first treatment (TWI), a participant can see after each iteration the actions (colour of the four rectangles) and the total earnings of herself and her neighbours (values in the rectangles). A participant's earnings are accumulated over four games (one with each neighbour) that each of them plays in a single round (figure 1a). Each TWI participant thus has the possibility to compare the performance of their neighbours with their own and use this information to change their behaviour, i.e. imitate (or not) the better ones. In the second treatment (TWO), they no longer get the information about the earnings of their neighbours accumulated in the previous round. They can only observe their neighbours' actions (figure 1b). As a consequence, participants need to use another mechanism to update their behaviour. If there are no differences between treatment TWI and TWO then this would mean that the earnings do not play a role in the decision-making process. If on the other hand a difference is observed, then one could argue that their future behaviour is determined by the success of their opponents and that they chose the action that would also give a high reward. In other words, they use some form of imitation when selecting one of the two actions.

In total, nine sessions were performed of the spatial iterated PD, involving a total of 144 participants. These participants consisted mostly of students of the Vrije Universiteit Brussel and the Université Libre de Bruxelles. Ethical approval for these experiments was obtained from the Ethical Commission for Human Sciences at the Vrije Universiteit Brussel (ECHW2015_3). Treatment TWI consisted of five sessions wherein 80 persons participated and treatment TWO consisted of four sessions with 64 participants. The participants were relatively balanced by gender, with 54% of the participants being male and 46% being female. The average age in both treatments was $24 \pm 5$ years (see the electronic supplementary material for gender and age details per treatment).

At the end of the experiment, the participants were paid according to their success in the games over the 50 rounds. The following exchange rate between virtual and real money was used: 1 point = 0.02 euro. Worst case, every neighbour defected while the focal player cooperated, which would produce a total gain of zero over all the rounds for the focal player. In the best case, all neighbours were cooperative and the central player defected, resulting in 1200 points or a 24 euro total gain for the focal player. Given that a high level of cooperation was expected (see above), the expected average gain was set in such a way that it would be close to the average hourly salary in Belgium. Apart from the gains they accumulated over the 50 rounds, the participants also received a 2.50 euro show-up fee.

The treatments were performed over a period of approximately three weeks, with each treatment session taking no more than 1 h. The experimental software was developed by the authors. Participants were able to perform the experiment in English, French or Dutch. At the beginning of the experiment, each participant read a detailed description of the experiment with a little test at the end to make sure that they understood the instructions (see the electronic supplementary material for tests and instructions). The examples in this test were chosen to be as neutral as possible. Furthermore, to make sure that the players were not framed by the examples themselves, we used two different sets, allowing us to verify the possibility of framing effects in the observed results. As can be observed in table 1, this appeared not to be the case.

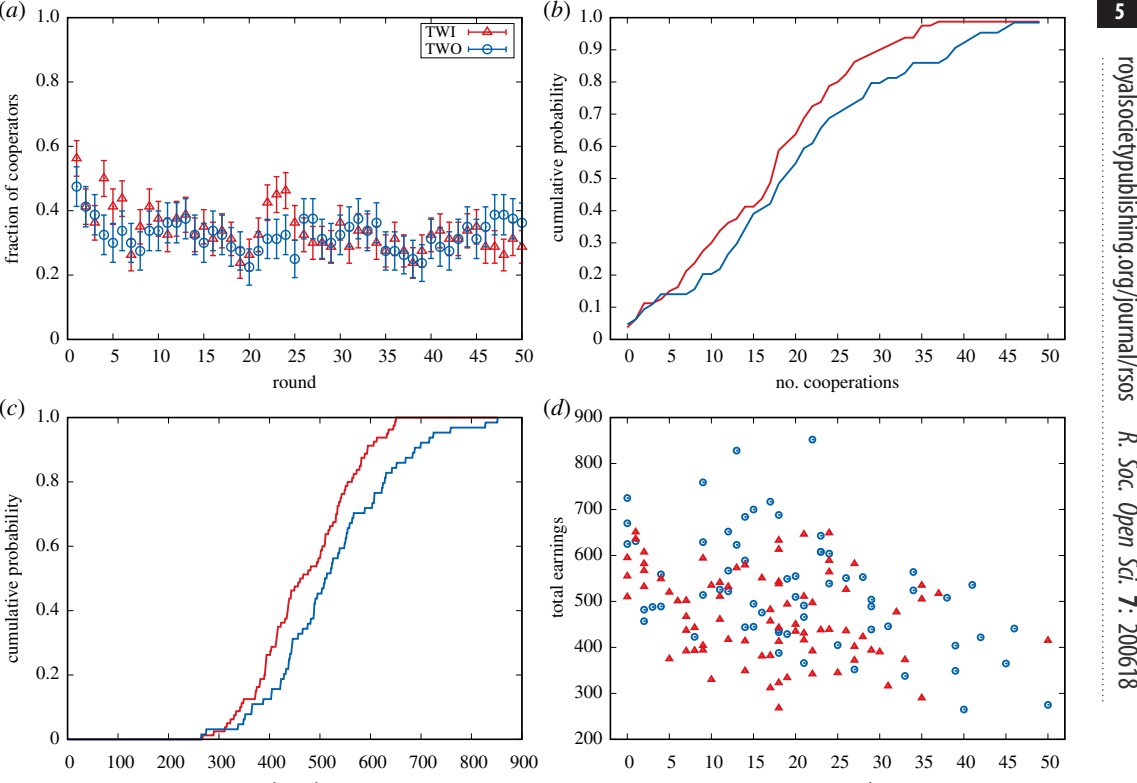

**Figure 2.** Analysis of level of cooperation and earnings in both treatments of the experiment. (*a*) The level of cooperation and standard deviations, where the latter is calculated as $\sqrt{p(1-p)/N}$, with $p$ the probability of cooperation in that round and $N$ the number of players. (*b*) Cumulative distribution of cooperation of the participants for treatment TWI and TWO. (*c*) Cumulative distribution of total earnings in the game for all participants in treatment TWI and TWO. (*d*) Focus on each participant in terms of the number of times they cooperated (*x*-axes) and how much they earned over the entire treatment (*y*-axes). Each symbol is a player, with circles corresponding to the participants that received only feedback about the actions of all neighbours in the previous round and triangles corresponding to participants receiving both feedback on the action and the success of the neighbouring players.

All statistical tests were performed using R. To test differences in the medians of two distributions (as in figure 3) a Welch *t*-test was used, to verify the claim made in [33]. Each decision made by a participant is considered to be a data point. To test whether full distributions are different, as in figure 2*b,c*, the two-sample Kolmogorov–Smirnoff (KS) test was used. The dependency of cooperation on the presence (or absence) of pay-off information in each round was assessed with a generalized linear model (GLM). In this model, the following variables were considered: the 'round' variable which indicates the round wherein the action occurred and an 'info' variable indicating whether the action was taken in TWI or TWO. In addition, we included a 'rnd' variable consisting of random values (0 or 1 with 50% probability), serving as a control feature and the variable 'example' which tells us which example people saw in the introduction part before staring the game session.

The motivation for adding the example variable is to ensure that the differences in pre-game tests that were performed by the participants did not influence the likelihood of cooperation. Before the start of the actual game, the participants read through a tutorial and their understanding was tested with a small quiz. Concretely they were asked to answer how much they would earn in a few situations, where each situation is defined by the actions taken by their neighbours and themselves. Such tests are essential to ensure that the players understand the instructions, as some research shows that almost 30% of participants will not carefully read the instructions which will decrease the statistical power of the results [36]. As there are many possible configurations of neighbour and personal choices, the test was limited to only a few of them, i.e. two sets of four examples were used to test the understanding (see the electronic supplementary material for details). By introducing the variable 'example', we can examine whether the cooperation levels in TWI and TWO are influenced by the tests the participants were exposed too.

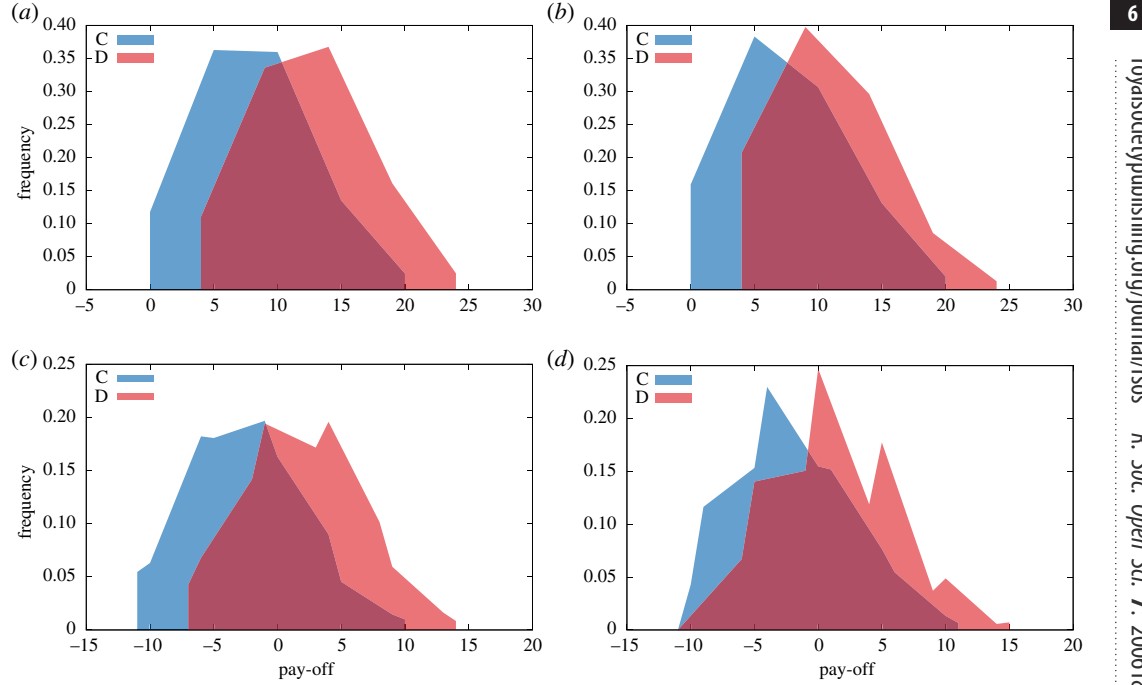

**Figure 3.** Comparison of actual (*a,b*) and relative (*c,d*) pay-offs when players cooperate and when they defect. (*a,c*) Treatment TWO and (*b,d*) treatment TWI. The pay-offs of defectors are significantly larger than the pay-offs of cooperators despite the fact that the cooperation condition was fulfilled.

## 3. Results

Figure 2 shows the global results for both treatments in terms of cooperation and earnings. The statistical analysis of these global indicators reveals that they are equivalent: the cooperation level in both cases starts from 50% cooperation and drops to around 30% after approximately 10 rounds (figure 2*a*). The standard deviations are largely overlapping. This equivalence is confirmed by a KS test (confidence level 0.05), producing *p*-values 0.6037 ($D = 0.12813$) and 0.127 ($D = 0.19688$) for the cooperation cumulative distribution (figure 2*b*) and earnings cumulative distribution (figure 2*c*), respectively.

As the level of cooperation (which is between 30% and 40%) does not meet the expectations formulated in Methods, we examine in figure 3 in more detail the distributions of acquired pay-offs for when participants cooperated (C) and when they defected (D). Based on [33], the hypothesis is that the C and D pay-offs are not significantly different, as this would explain the lower observed cooperation levels for a regime where $Q^* > k$. As can be observed, we see the opposite: defectors earn significantly more than cooperators in both treatments, both when considering absolute and relative pay-offs (see statistical test in table 2). Our results appear thus to not lead to the same conclusions as those suggested by Rand *et al.* [33].

Focusing on the individual cooperativeness, i.e. the number of times a person cooperated, and the individual earnings over the entire treatment (see figure 2*d*), one can observe that the data for this global indicator are largely overlapping. Yet, qualitatively there appears to be more variation in the case where players were only informed about the actions of the neighbours in the previous round (i.e. TWO). This variation in the behaviour can be quantified by considering the standard error within both treatments, as visualized in figure 4. One can directly observe that the standard error in TWO remains constant over time, meaning that there appears to be no learning/convergence in the actions selected by the participants. However, in TWI, the standard error appears to decrease. So over time, participants in the TWI treatment are less exploratory in their actions than participants in TWO.

In order to grasp more clearly the difference in decision-making between treatments TWI and TWO observed in figure 4, we statistically analyse the likelihood to cooperate at each round using a GLM (see Methods). Next to the essential 'round' and 'info' variables two sanity check variables, i.e. 'rnd' and 'example', were introduced. The latter assure that the observations are not simply random or associated with the tests participants performed prior to the experiments themselves.

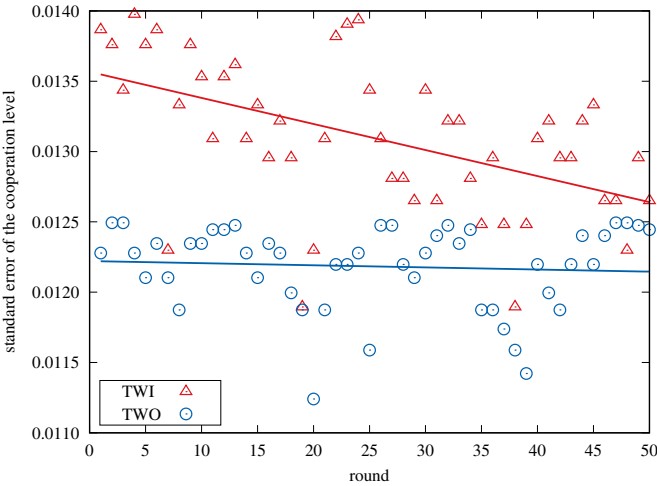

**Figure 4.** Standard error of the level of cooperation for TWO and TWI. Whereas the standard error remains more or less constant over the 50 rounds in the TWO treatment, the error appears to decrease for TWI.

**Table 2.** Welch's $t$-test for evaluating the significance in the absolute and relative pay-off differences for cooperative and defective decisions.

| treatments | absolute pay-offs | | relative pay-offs | |
|---|---|---|---|---|
| | Welch's $t$ statistics | $p$-value | Welch's $t$ statistics | $p$-value |
| TWO | −25.32 | <0.0001*** | −25.98 | <0.0001*** |
| TWI | −19.62 | <0.0001*** | −20.40 | <0.0001*** |

The GLM results shown in table 1 immediately reveal that the variable 'example' does not affect the action choice and that only round and info are significantly associated with the level of cooperation in both treatments. Moreover, the control variable rnd does not predict the actions selected by participants, which is as it should be. The GLM analysis results can be summarized as follows: (i) an increase in rounds leads to a reduction in cooperation, and (ii) providing information about pay-offs affects the decision process because knowing the pay-off of the neighbours introduces a reduction in the cooperativeness of the participants.

To get at the heart of the matter, the relationship between the pay-off difference experienced by each participant and the probability to switch between cooperation and defection (or vice versa) is visualized in figure 5, an analysis similar to [20,23]. When the pay-off differences are not large, no matter the action, there is almost no difference in changing the behaviour between both treatments. However, once the difference becomes big enough a clear signal emerges: providing information about the success of neighbours leads to a stronger response, leading to a higher probability to imitate defective behaviour or to maintain it at a cost of lowering the overall benefits that can be reaped in the game. Interestingly, the absence of pay-off information, which does not allow for comparisons of success, produces similar probabilities of switching from C to D (and from D to C). A GLM relating the probability of changing the action to the differences in pay-offs (diff), the treatment (info) and combination of both variables (diff × info) reveals clearly a predictive correlation with the difference in pay-off, and the treatment when it is combined with those pay-off differences (table 3). Every unit increase in diff leads to a significant increase in the probability of changing the action, and this for this variable alone or in combination with the variable info that expresses the treatment wherein the decision was made.

The same data can be used to examine the relationship between the number of neighbouring cooperators and the probability of cooperation, as is visualized in figure 6. This result as well as the GLM in table 4 reveals the same signature as in the Moody conditional cooperation papers [21,23]. Upon closer examination of figure 6 for the situation where there are either one or three cooperative neighbours, one can observe differences in the likelihood of cooperation between the treatments TWO

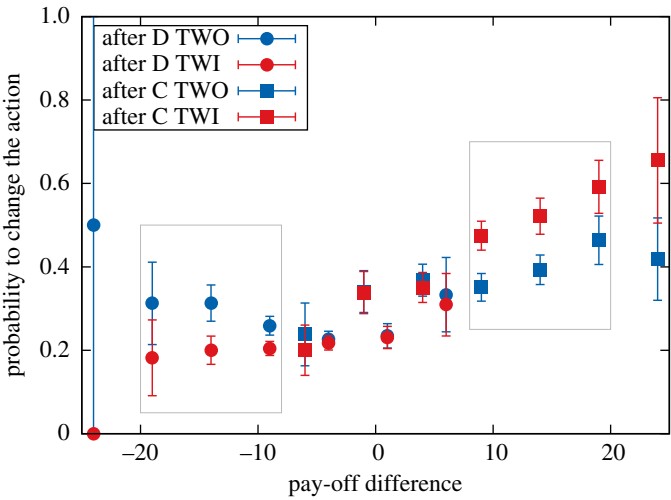

**Figure 5.** Correlation plot between pay-off difference and the probability to switch action given a specific prior action. Blue corresponds to the TWO treatment and red refers to the TWI treatment. The large light grey rectangles highlight the difference between both treatments: the right rectangle reveals that when information about the pay-off difference with the neighbours is available and after playing C, the focal player is more probable to switch behaviour, i.e. play D. The left rectangle shows the opposite, i.e. when pay-offs can be compared the focal player is likely to change after playing D. The choice to play C or D is thus influenced by the presence of pay-off information.

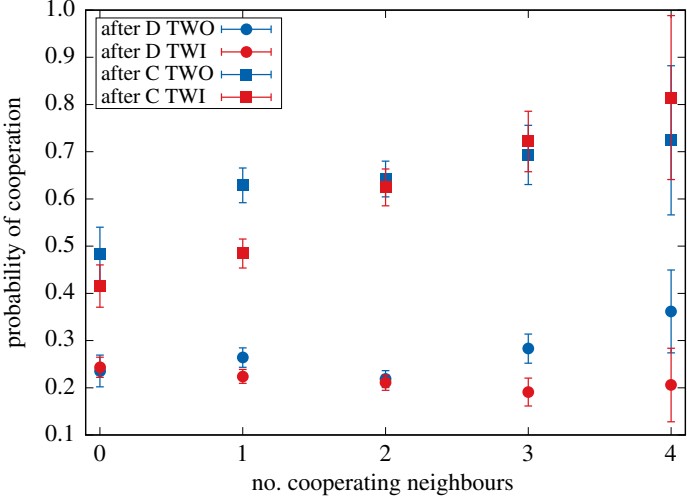

**Figure 6.** Moody conditional cooperator behaviour as observed in both treatments. Note the strong differences between both treatments for 1 (in case of playing C before) and 3 (in case of playing D before) cooperative neighbours.

**Table 3.** Generalized linear model for the probability to change the action in both treatments (shown in figure 5) (glm(formula = change ∼ diff + info + rnd + diff × info, family = binomial(`logit`))). (The significant variables are (i) 'diff' revealing that the probability of changing actions is correlated to the difference in pay-offs with the neighbours, and (ii) the combination 'diff:info', telling us that this correlation is different in treatments with (TWI) and without (TWO) information.)

| variable | estimate | s.e. | z-value | Pr(|z|) |
|---|---|---|---|---|
| (intercept) | −0.879433 | 0.050016 | −17.583 | $<2 \times 10^{-16***}$ |
| diff | 0.026458 | 0.004220 | 6.269 | $3.63 \times 10^{-10***}$ |
| info | −0.012179 | 0.057056 | −0.213 | 0.831 |
| rnd | 0.043624 | 0.056190 | 0.776 | 0.438 |
| diff : info | 0.037928 | 0.006013 | 6.308 | $2.83 \times 10^{-10***}$ |

**Table 4.** Generalized linear model for the probability of cooperation depending on the previous action of the focal player and the cooperation level in the neighbourhood in both treatments (shown in figure 6) (lm( formula = action $\sim$ info + $N\_$coop + prev_action)). (The variables are all significant: (i) 'info' is once again confirming the difference of behaviour between the two treatments with TWI and without (TWO) information, (ii) '$N\_$coop' is the number of cooperating neighbours of the focal player in the previous round, and (iii) 'prev_action' is the action of the focal player in the previous round.)

| variable | estimate | s.e. | z-value | Pr(\|z\|) |
|---|---|---|---|---|
| (intercept) | 0.215331 | 0.012826 | 16.789 | $<2 \times 10^{-16***}$ |
| info | −0.037874 | 0.010861 | −3.487 | $0.000491^{***}$ |
| N_coop | 0.027424 | 0.005606 | 4.892 | $1.02 \times 10^{-06***}$ |
| prev_ action | 0.352350 | 0.011089 | 31.775 | $<2 \times 10^{-16***}$ |

and TWI. In both cases, participants are less likely to switch to cooperative behaviour. Yet the data in figure 5 and the statistical analysis in table 3 are needed to reveal more clearly what is happening, suggesting that the behaviours produced by both update mechanisms are difficult to distinguish.

# 4. Discussion

Do humans imitate more successful neighbours in networks? Do pay-off differences matter in the decision to cooperate or defect? In the current work, an experiment consisting of two treatments was performed to answer these questions as an earlier experiment in a networked game did not provide support for imitation [26]. The results described in detail in the previous section show that, although on a global scale there is no difference in levels of cooperation or earnings for the entire network, the knowledge of the neighbours' pay-off influences the focal player's choice to cooperate or defect, and this in a manner that appears to be consistent with the mechanism of 'imitation of the more successful'. As was visualized in figure 5 and statistically supported by the GLM results in table 3, the probability of changing to the action of the successful neighbour is more likely when that neighbour earns more than the focal player and the focal player is made aware of that fact.

Notwithstanding these results, it is, of course, impossible to claim that each participant in the experiment used indeed exactly this procedure to choose between cooperation and defection. The fact that a participant knows the success of her neighbours does not mean that she will use it to imitate the action of the most successful one. The differences between both treatments might simply be a consequence of a stronger emotional response induced by the pay-off difference and not a 'rational' choice to act like the more successful neighbours. Another possible explanation is that the speed of the learning dynamics is faster in the treatments with less information. However, given the length of the experiment and the fact that there is no difference in the level of cooperation over time in the two treatments, this hypothesis is less likely to be true. Nonetheless, the evidence provided here shows at least that providing insight into the success of the neighbours influences how people decide to act. Our results thus contradict Kirchkamp & Nagel [26], revealing that the presence of pay-off information leads to imitation in a networked PD and not the type of information, i.e. actual versus average information. Moreover, as in Apesteguia *et al.* [32], the bigger the differences in pay-off, the stronger the effect. Finally, the results suggest that an update mechanism that employs comparisons between pay-offs, as, for instance, the Fermi pairwise comparison rule [15,37], is more likely to be related to the observed behaviour, providing explicit boundaries on which update mechanisms are more meaningful for simulating human decision-making.

Although the current analysis focuses on the average behaviour, further studies can be performed on how individuals respond in the experiment, aiming to reveal behavioural types. These types should not only consider the responses towards observations but also the frequency with which these observations are made, as human behaviours are influenced by the context they find themselves in. This work is currently in progress and will be part of an independent manuscript.

In conclusion, the experiment discussed here shows that, given access to the information, people switch their behaviour when their neighbours are more successful, corresponding to the notion of imitation that has been frequently used in theoretical models studying the evolution of cooperation [9]. This evidence is impossible to derive from the cooperation level, requiring an in depth analysis of the participants' behaviour. Previous experimental situations were either negative [21,26] or showed only a very weak

relevance [24]. It should be noted that the suggested criteria for the promotion of cooperation in networks [33] may require a revision, as the anticipated cooperation level for the given condition was not observed. It remains a question then how connectivity in the network leads to more pro-social behaviour. Presumably bigger differences between degree and the benefit-to-cost ratio need to be considered. As a corollary, one may wonder what influence a stronger difference may have on the appearance of imitating behaviour. For now, the results and analyses provided here revealed explicitly that knowledge about the success of others affects how we change behaviour in spatial social dilemmas like the PD.

Ethics. Ethical approval by reference number ECHW2015_3 was obtained from the Ethical Commission for Human Sciences at the Vrije Universiteit Brussel to perform this experiment. All the participants in the study had to give their informed consent prior to the participation.

Data accessibility. The data are available at Dryad Digital Repository: https://doi.org/10.5061/dryad.7fq25s8 [38].

Authors' contributions. J.G. designed, performed the experiments; J.G. and T.L. analysed the results and discussed them; J.G. and T.L. wrote the manuscript.

Competing interests. The authors declare no conflict of interest. The funding agency had no role in the design of the study; in the collection, analyses or interpretation of data; in the writing of the manuscript, or in the decision to publish the results.

Funding. J.G. is funded by FWO - Research Foundation Flanders. T.L. is supported by the Fondation de la Recherche Scientifique (F.R.S.-FRNS) through the project PDR 31257234. T.L. is supported by the FuturICT2.0 (www.futurict2. eu) project funded by the FLAG-ERA Joint Transnational call (JTC) 2016.

Acknowledgements. J.G. and T.L. thank Pieter Libin and Nathaniel Mon Père for their valuable comments on the manuscript. We also thank all the students for participating in our experiment.

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
