## [Reviewer comments · Royal Society Open Science]

Review History

RSOS-191365.R0 (Original submission)

Review form: Reviewer 1

Is the manuscript scientifically sound in its present form?

Yes

Are the interpretations and conclusions justified by the results?

Yes

Is the language acceptable?

No

Do you have any ethical concerns with this paper?

Yes

Have you any concerns about statistical analyses in this paper?

No

Recommendation?

Major revision is needed (please make suggestions in comments)

Comments to the Author(s)

The authors report an experiment exploring whether people imitate better performing neighbours when playing an iterated prisoner's dilemma on a lattice. To answer this question, the authors conduct a two-treatment experiment, where participants, after each round, are informed vs not-informed about the payoff of the neighbours. The main result of the paper is that, although there are no statistically significant differences between the levels of cooperation, there are differences in how people update their strategy, and these differences are well explained by the payoff difference between the focal player and the best performing neighbour. In sum, these results provide a piece of evidence in support of the assumption that people do update their strategy according to the "imitate the best" rule.

I am sympathetic towards this paper, which I think can make a useful and important contribution to the literature. At the same time, I think that the Methods is missing some key information. Moreover, I have found a terrible number of typos and grammar mistakes. I was very surprised by this, especially because this is already a revision. I would strongly recommend the authors to carefully revise their writing before resubmitting.

I list below some specific comments. They mainly regard the methods section.

Methods

- How many rounds? (I see that this information is given at the end of the methods section. I don't think that this is the right place for such an important information, which should be given much earlier)
- The stakes in the PD seem very small compared to the participation fee. Are you not concerned that this makes people to not take the game seriously?
- What is the need to include a time pressure? Note that previous work has shown that time pressure can affect cooperation, generally in a positive way (Rand, Greene and Nowak, 2012; Rand, 2016), although there is also some work arguing that extreme time pressure may undermine cooperation (Capraro & Cococcioni, 2016). How do your results relate to this line of work?
- Moreover, from the text it is not clear what was the time constraint. How many seconds did participants have to make their choice?
- How did you determine the end of the game? Random? Pre-determined?
- My understanding is that different treatments were conducted in different times. This is a violation of the random assignment procedure. Why do you think it is acceptable in this case? Moreover, why do you have 5 session with information but only four without? How was the sample size determined? Was it pre-determined?

Results

- The sentence "Before the start of the actual game, the participants read through a tutorial and their understanding is tested with a small quiz. Concretely they are asked to answer how much they earn in a few situations, where each situation is defined by the actions taken by their neighbors and themselves." Should not be in the Results section, but in the methods.

Discussion

- "The differences between both treatments might simply be a consequence of a stronger emotional response induced by the payoff difference and not a "rational" choice to act like the more successful neighbors": good point, especially because participants act under time pressure, and this favours emotional responses over deliberative ones.

Review form: Reviewer 2

Is the manuscript scientifically sound in its present form?

No

Are the interpretations and conclusions justified by the results?

No

Is the language acceptable?

No

Do you have any ethical concerns with this paper?

No

Have you any concerns about statistical analyses in this paper?

Yes

Recommendation?

Reject

Comments to the Author(s)

The manuscript reports an experimental study on the behaviour of people when making decisions in a Prisoner's Dilemma (PD) game played on a squared lattice network. The main research question is to measure whether participants take decisions imitating their most successful neighbour, i.e. the neighbour who earned the most in the previous round of the game. The authors implemented two experimental treatments, involving a total of 144 subjects, where the information about neighbours' previous payoff was either available or hidden, while neighbours' actions were always shown. Reported results depict that global cooperation levels are similar for both treatments while average earnings differ sensibly for cooperators and defectors. In the spirit of their research question, it appears that payoff differences influence the probability of changing action. Also, it is shown that probabilities to switch to another action are correlated to the number of cooperators in participants' neighbourhoods, in agreement with the previously observed behaviour of moody conditional cooperators.

The research topic is of interest and widely debated in the related literature, which is duly reported. However, the presentation, the statistical analysis, the presented results and the structure of the paper are largely below the threshold for considering the work as publishable. All text seems to be in a draft version: many grammar mistakes, typos, unfinished/unclear or extremely long sentences. The statistical analysis is not properly carried out: it looks like the authors used all observations, i.e. individual actions, as independent ones, while they actually have only 5 vs. 4 independent observations, i.e. the number of independent groups. Some results are meaningless for the main research question, which only arrives at Figure 5, and many more specific ones should be done to better, or properly, answer it. Finally, the structure should be strongly revised: KS test is unnecessarily explained, Figure 3 explanation comes before the one of Fig. 2d, Figure 6 and Table 4 are only mentioned in the discussion, the example variable explanation should not be in the results section, specific results in figures and tables should not be commented in the discussion. For these reasons, although I found the experimental procedure properly done and robust and the leading research question adequate, the manuscript cannot be considered for publication in its present version and a future re-submission is suggested.

Here there are some more specific comments to improve the quality of the work:

- A deep revision of the text is required, I cannot list all the errors. Pages 3-4 of the Introduction and Methods section are particularly problematic.

- Treatment names are introduced only in section 3 while they are already mentioned in the previous section.
- It is either prisoner's dilemma or prisoners' dilemma.
- The definition of Q^* in the Introduction is too technical and it should be moved to the methods/results, if it is really needed. However, it does not add much to the specific research question. It is not specified which is the used topology there.
- The KS test may be not enough or adequate for a complete statistical analysis. I would recommend to use the Mann-Whitney test too/instead. However, as I mentioned before, the number of independent observations is 9 and the statistical analysis on cooperation levels and earnings, if really needed, should be done with them, considering average group levels during the entire experiment. Having p-values of the order of 10^{-133} with 144 subjects is simply impossible. All tables and GLM have the same issue.
- Figure 4 results are unclear. It is not clear which observation was discarded, and how. Again, instead than SD, error bars should be plotted, considering group averages as independent observations and normalising according to them, i.e. dividing by 4 or 5, if needed.
- The example variable is a good sanity check that can be moved to the supplementary material and just mentioned in the main manuscript. The same applies to the random variable, if needed.
- in the Appendix, what do you mean with "balanced participants"? I guess they were gender-balanced, or is it another kind of balance? In any case, more specific statistics on participants by treatment should be included in a dedicated section of the supplementary material, as well as the days of the experiments.
- references have to be revised, such as 27 or 29. There is not a standardised style and all titles are missing, making their search quite difficult.
- finally, I don't understand why the authors focused their study on average values instead than on individual behaviours; especially in an experimental study where the number of subjects is very small. An analysis on specific behavioural types of participants could have been much more intriguing and adequate than looking at, sometimes tiny, average differences between treatments. It could be the case that the behaviour of imitating the most successful neighbour exists, even to a larger extent, only for a certain percentage of subjects, as well as for the moody conditional behaviour.

Decision letter (RSOS-191365.R0)

28-Sep-2019

Dear Dr Grujic:

Manuscript ID RSOS-191365 entitled "Do people imitate when making decisions? - evidence from a spatial prisoners dilemma experiment" which you submitted to Royal Society Open Science, has been reviewed. The comments from reviewers are included at the bottom of this letter.

In view of the criticisms of the reviewers, the manuscript has been rejected in its current form. However, a new manuscript may be submitted which takes into consideration these comments.

Please note that resubmitting your manuscript does not guarantee eventual acceptance, and that your resubmission will be subject to peer review before a decision is made.

Your resubmitted manuscript should be submitted by 27-Mar-2020. If you are unable to submit by this date please contact the Editorial Office.

on behalf of Professor Matjaz Perc (Associate Editor) and Essi Viding (Subject Editor)
openscience@royalsociety.org

Reviewers' Comments to Author:

Reviewer: 1

Comments to the Author(s)

The authors report an experiment exploring whether people imitate better performing neighbours when playing an iterated prisoner's dilemma on a lattice. To answer this question, the authors conduct a two-treatment experiment, where participants, after each round, are informed vs not-informed about the payoff of the neighbours. The main result of the paper is that, although there are no statistically significant differences between the levels of cooperation, there are differences in how people update their strategy, and these differences are well explained by the payoff difference between the focal player and the best performing neighbour. In sum, these results provide a piece of evidence in support of the assumption that people do update their strategy according to the "imitate the best" rule.

I am sympathetic towards this paper, which I think can make a useful and important contribution to the literature. At the same time, I think that the Methods is missing some key information. Moreover, I have found a terrible number of typos and grammar mistakes. I was very surprised by this, especially because this is already a revision. I would strongly recommend the authors to carefully revise their writing before resubmitting.

I list below some specific comments. They mainly regard the methods section.

Methods

- How many rounds? (I see that this information is given at the end of the methods section. I don't think that this is the right place for such an important information, which should be given much earlier)
- The stakes in the PD seem very small compared to the participation fee. Are you not concerned that this makes people to not take the game seriously?

- What is the need to include a time pressure? Note that previous work has shown that time pressure can affect cooperation, generally in a positive way (Rand, Greene and Nowak, 2012; Rand, 2016), although there is also some work arguing that extreme time pressure may undermine cooperation (Capraro & Cococcioni, 2016). How do your results relate to this line of work?
- Moreover, from the text it is not clear what was the time constraint. How many seconds did participants have to make their choice?
- How did you determine the end of the game? Random? Pre-determined?
- My understanding is that different treatments were conducted in different times. This is a violation of the random assignment procedure. Why do you think it is acceptable in this case? Moreover, why do you have 5 session with information but only four without? How was the sample size determined? Was it pre-determined?

Results

- The sentence "Before the start of the actual game, the participants read through a tutorial and their understanding is tested with a small quiz. Concretely they are asked to answer how much they earn in a few situations, where each situation is defined by the actions taken by their neighbors and themselves." Should not be in the Results section, but in the methods.

Discussion

- "The differences between both treatments might simply be a consequence of a stronger emotional response induced by the payoff difference and not a "rational" choice to act like the more successful neighbors": good point, especially because participants act under time pressure, and this favours emotional responses over deliberative ones.

Reviewer: 2

Comments to the Author(s)

The manuscript reports an experimental study on the behaviour of people when making decisions in a Prisoner's Dilemma (PD) game played on a squared lattice network. The main research question is to measure whether participants take decisions imitating their most successful neighbour, i.e. the neighbour who earned the most in the previous round of the game. The authors implemented two experimental treatments, involving a total of 144 subjects, where the information about neighbours' previous payoff was either available or hidden, while neighbours' actions were always shown. Reported results depict that global cooperation levels are similar for both treatments while average earnings differ sensibly for cooperators and defectors. In the spirit of their research question, it appears that payoff differences influence the probability of changing action. Also, it is shown that probabilities to switch to another action are correlated to the number of cooperators in participants' neighbourhoods, in agreement with the previously observed behaviour of moody conditional cooperators.

The research topic is of interest and widely debated in the related literature, which is duly reported. However, the presentation, the statistical analysis, the presented results and the structure of the paper are largely below the threshold for considering the work as publishable. All text seems to be in a draft version: many grammar mistakes, typos, unfinished/unclear or extremely long sentences. The statistical analysis is not properly carried out: it looks like the authors used all observations, i.e. individual actions, as independent ones, while they actually have only 5 vs. 4 independent observations, i.e. the number of independent groups. Some results are meaningless for the main research question, which only arrives at Figure 5, and many more specific ones should be done to better, or properly, answer it. Finally, the structure should be strongly revised: KS test is unnecessarily explained, Figure 3 explanation comes before the one of Fig. 2d, Figure 6 and Table 4 are only mentioned in the discussion, the example variable explanation should not be in the results section, specific results in figures and tables should not

be commented in the discussion. For these reasons, although I found the experimental procedure properly done and robust and the leading research question adequate, the manuscript cannot be considered for publication in its present version and a future re-submission is suggested.

Here there are some more specific comments to improve the quality of the work:

- A deep revision of the text is required, I cannot list all the errors. Pages 3-4 of the Introduction and Methods section are particularly problematic.
- Treatment names are introduced only in section 3 while they are already mentioned in the previous section.
- It is either prisoner's dilemma or prisoners' dilemma.
- The definition of Q^* in the Introduction is too technical and it should be moved to the methods/results, if it is really needed. However, it does not add much to the specific research question. It is not specified which is the used topology there.
- The KS test may be not enough or adequate for a complete statistical analysis. I would recommend to use the Mann-Whitney test too/instead. However, as I mentioned before, the number of independent observations is 9 and the statistical analysis on cooperation levels and earnings, if really needed, should be done with them, considering average group levels during the entire experiment. Having p-values of the order of $10^{-(133)}$ with 144 subjects is simply impossible. All tables and GLM have the same issue.
- Figure 4 results are unclear. It is not clear which observation was discarded, and how. Again, instead than SD, error bars should be plotted, considering group averages as independent observations and normalising according to them, i.e. dividing by 4 or 5, if needed.
- The example variable is a good sanity check that can be moved to the supplementary material and just mentioned in the main manuscript. The same applies to the random variable, if needed.
- in the Appendix, what do you mean with "balanced participants"? I guess they were gender-balanced, or is it another kind of balance? In any case, more specific statistics on participants by treatment should be included in a dedicated section of the supplementary material, as well as the days of the experiments.
- references have to be revised, such as 27 or 29. There is not a standardised style and all titles are missing, making their search quite difficult.
- finally, I don't understand why the authors focused their study on average values instead than on individual behaviours; especially in an experimental study where the number of subjects is very small. An analysis on specific behavioural types of participants could have been much more intriguing and adequate than looking at, sometimes tiny, average differences between treatments. It could be the case that the behaviour of imitating the most successful neighbour exists, even to a larger extent, only for a certain percentage of subjects, as well as for the moody conditional behaviour.

Author's Response to Decision Letter for (RSOS-191365.R0)

See Appendix A.

RSOS-200618.R0

Review form: Reviewer 1

Is the manuscript scientifically sound in its present form?

Yes

Are the interpretations and conclusions justified by the results?

Yes

Is the language acceptable?

Yes

Do you have any ethical concerns with this paper?

No

Have you any concerns about statistical analyses in this paper?

Yes

Recommendation?

Accept with minor revision (please list in comments)

Comments to the Author(s)

I thank the authors for addressing my comments. I think that the paper has been definitely improved. At the same time, I am not convinced by they way my comment regarding the risk of inducing time pressure has been addressed. The authors' response is that inducing people to respond within 30 seconds does not induce time pressure because other experiments used 30 seconds to induce time delay. This logic does not work because it fundamentally misses the point of time pressure vs time delay techniques. The number of seconds is not really the point of these techniques. The point is the feeling that participants sense when they have to answer **within** a certain amount of time vs **after** a certain amount of time. In one case they feel under pressure, in the other case they feel relaxed. Of course, the extent to which they feel under pressure may vary; so a time pressure of 10 seconds is **quantitatively** stronger than a time pressure of 30 seconds. However, also 30 seconds is **qualitatively** a time pressure. Therefore, I think that the authors have to re-address this point in a better way.

(En passant, let me note that it remains unclear to me why the authors introduced a time pressure in their experiment. To me, there was no need. But perhaps I'm misunderstanding something.)

Review form: Reviewer 2

Is the manuscript scientifically sound in its present form?

Yes

Are the interpretations and conclusions justified by the results?

Yes

Is the language acceptable?

Yes

Do you have any ethical concerns with this paper?

Yes

Have you any concerns about statistical analyses in this paper?

No

Recommendation?

Accept with minor revision (please list in comments)

Comments to the Author(s)

The manuscript, with respect to its previous version, largely improved from all points of view. All the introductory part is now much clearer and results, as well as their discussion, have been polished. Also, the authors satisfactorily addressed most of my previous comments. I am still a bit skeptical about the use of all observations as independent ones but I guess it is the best they can do. The paper still provides important evidences, perhaps not statistically significant as they state, to the research topic and it can be now considered for publication after addressing the few minor remarks reported below.

- Since the authors decided to not implement Wilcoxon test, it is unnecessary to talk about it in the Methods section.

- The axes in Figures 3 should be the same to encourage the reader to a proper comparison. At least, when considering pairs of row panels.

- The authors should state what they intend as independent observation, i.e. individual action, in the Methods section and I suggest them to get rid of those fishy p-values just stating something like $p < .0001$ instead.

- The authors could mention in the conclusions more insightful remarks for a follow-up research on the related topic as the framework of behavioral types or, generally speaking, a more detailed analysis that can be done at the individual level. What they found looking at the "average" behavior can be largely heterogeneous at the individual one; as it is already visible in their work but it is not part of the main research question.

Decision letter (RSOS-200618.R0)

Dear Dr Grujic

On behalf of the Editor, I am pleased to inform you that your Manuscript RSOS-200618 entitled "Do people imitate when making decisions? - evidence from a spatial prisoners dilemma experiment" has been accepted for publication in Royal Society Open Science subject to minor revision in accordance with the referee suggestions. Please find the referees' comments at the end of this email.

The reviewers and Subject Editor have recommended publication, but also suggest some minor revisions to your manuscript. Therefore, I invite you to respond to the comments and revise your manuscript.

- Ethics statement

If your study uses humans or animals please include details of the ethical approval received, including the name of the committee that granted approval. For human studies please also detail

whether informed consent was obtained. For field studies on animals please include details of all permissions, licences and/or approvals granted to carry out the fieldwork.

- Data accessibility

If you wish to submit your supporting data or code to Dryad (<http://datadryad.org/>), or modify your current submission to dryad, please use the following link:
<http://datadryad.org/submit?journalID=RSOS&manu=RSOS-200618>

- Competing interests

- Authors' contributions

- Acknowledgements

- Funding statement

Because the schedule for publication is very tight, it is a condition of publication that you submit the revised version of your manuscript before 14-Jun-2020. Please note that the revision deadline will expire at 00.00am on this date. If you do not think you will be able to meet this date please let me know immediately.

on behalf of Professor Matjaz Perc (Associate Editor) and Essi Viding (Subject Editor)
openscience@royalsociety.org

Reviewer comments to Author:
Reviewer: 1

Comments to the Author(s)
I thank the authors for addressing my comments. I think that the paper has been definitely improved. At the same time, I am not convinced by they way my comment regarding the risk of

inducing time pressure has been addressed. The authors' response is that inducing people to respond within 30 seconds does not induce time pressure because other experiments used 30 seconds to induce time delay. This logic does not work because it fundamentally misses the point of time pressure vs time delay techniques. The number of seconds is not really the point of these techniques. The point is the feeling that participants sense when they have to answer *within* a certain amount of time vs *after* a certain amount of time. In one case they feel under pressure, in the other case they feel relaxed. Of course, the extent to which they feel under pressure may vary; so a time pressure of 10 seconds is *quantitatively* stronger than a time pressure of 30 seconds. However, also 30 seconds is *qualitatively* a time pressure. Therefore, I think that the authors have to re-address this point in a better way.
(En passant, let me note that it remains unclear to me why the authors introduced a time pressure in their experiment. To me, there was no need. But perhaps I'm misunderstanding something.)

Reviewer: 2

Comments to the Author(s)

The manuscript, with respect to its previous version, largely improved from all points of view. All the introductory part is now much clearer and results, as well as their discussion, have been polished. Also, the authors satisfactorily addressed most of my previous comments. I am still a bit skeptical about the use of all observations as independent ones but I guess it is the best they can do. The paper still provides important evidences, perhaps not statistically significant as they state, to the research topic and it can be now considered for publication after addressing the few minor remarks reported below.

- Since the authors decided to not implement Wilcoxon test, it is unnecessary to talk about it in the Methods section.
- The axes in Figures 3 should be the same to encourage the reader to a proper comparison. At least, when considering pairs of row panels.
- The authors should state what they intend as independent observation, i.e. individual action, in the Methods section and I suggest them to get rid of those fishy p-values just stating something like $p < .0001$ instead.
- The authors could mention in the conclusions more insightful remarks for a follow-up research on the related topic as the framework of behavioral types or, generally speaking, a more detailed analysis that can be done at the individual level. What they found looking at the "average" behavior can be largely heterogeneous at the individual one; as it is already visible in their work but it is not part of the main research question.

Author's Response to Decision Letter for (RSOS-200618.R0)

See Appendix B.

Decision letter (RSOS-200618.R1)

Dear Dr Grujic,

It is a pleasure to accept your manuscript entitled "Do people imitate when making decisions? - evidence from a spatial prisoner's dilemma experiment" in its current form for publication in Royal Society Open Science. The comments of the reviewer(s) who reviewed your manuscript are included at the foot of this letter.

on behalf of Professor Matjaz Perc (Associate Editor) and Essi Viding (Subject Editor)
openscience@royalsociety.org

Associate Editor Comments to Author (Professor Matjaz Perc):

Comments to the Author:

Thank you for the comprehensive revision of your manuscript, which we are happy to accept for publication in Royal Society Open Science.

Appendix A

Response to reviewers

Thank you for giving us the opportunity to resubmit our work. We hope that this thorough revision of the manuscript resolves all issues raised by both reviewers. All changes (additions/corrections) are highlighted in red in the new version of the manuscript. Please find below the answers to your comments and concerns.

Reviewer: 1

Comments to the Author(s)

The authors report an experiment exploring whether people imitate better performing neighbours when playing an iterated prisoner's dilemma on a lattice. To answer this question, the authors conduct a two-treatment experiment, where participants, after each round, are informed vs not-informed about the payoff of the neighbours. The main result of the paper is that, although there are no statistically significant differences between the levels of cooperation, there are differences in how people update their strategy, and these differences are well explained by the payoff difference between the focal player and the best performing neighbour. In sum, these results provide a piece of evidence in support of the assumption that people do update their strategy according to the "imitate the best" rule.

I am sympathetic towards this paper, which I think can make a useful and important contribution to the literature. At the same time, I think that the Methods is missing some key information. Moreover, I have found a terrible number of typos and grammar mistakes. I was very surprised by this, especially because this is already a revision. I would strongly recommend the authors to carefully revise their writing before resubmitting.

We thank the reviewer for her/his appreciation of our work. We apologize for the numerous typos, which should now be corrected in this new version. Please find the answers to your questions below. Your input has allowed us to substantially improve the quality of our manuscript.

I list below some specific comments. They mainly regard the methods section.

Methods

- How many rounds? (I see that this information is given at the end of the methods section. I don't think that this is the right place for such an important information, which should be given much earlier)

Indeed, 50 rounds were performed iteratively, as was specified at the end of the Methods section. We have now moved this all the way to the beginning, as requested. The number of rounds was chosen to avoid transient effects within experiments that do not last long enough. It has been

observed for instance that too short experiments cannot capture a potential increase in cooperation.

- The stakes in the PD seem very small compared to the participation fee. Are you not concerned that this makes people to not take the game seriously?

We apologize if this was not clear. As can be observed in Figure 2(d) most participants earned between 300 and 800 points over the 50 rounds of the game. As a consequence, they earned between 6 and 16 euros, which is still higher than the show-up fee. We have now added this information in the Methods section when discussing the exchange rate. The fact that the earnings are on the low side is also a consequence of the limited amount of cooperation that was finally observed. As explained in the paper, we were aiming for higher levels of cooperation using the theoretical conclusions drawn in the Rand et al paper. However, as we show, their conclusions appear not to hold in our data, lowering at the same time the gains of the participants. So they could have gained much more by cooperating, which is not what we observed.

The following text was added: "Worst case, every neighbor defects while the focal player cooperates, which would produce a total gain of zero over all the rounds. In the best case all neighbors are cooperative and the central player defects, resulting in \$1200\$ points or \$24\$ euro total gain for the focal player. Given that a high level of cooperation is expected (see above), the expected average gain was set in such a way that it would be close to the average hourly salary in Belgium."

- What is the need to include a time pressure? Note that previous work has shown that time pressure can affect cooperation, generally in a positive way (Rand, Greene and Nowak, 2012; Rand, 2016), although there is also some work arguing that extreme time pressure may undermine cooperation (Capraro & Cococcioni, 2016). How do your results relate to this line of work?

The participants were urged to decide within 30 seconds, yet, even when this time expired, they could still delay their choice. They just received a warning on the screen. This was made very clear to them in the instructions they received before the game. We informed them that the only consequence of them not playing in time is that the whole experiment could take too long. Therefore, there was no actual time pressure in the game.

Please, notice that in the papers cited above the time pressure was considered to make them answer in less than 10 seconds and not 30 seconds, and asking them to play in more than 30 seconds was considered a time delay. Thirty seconds is therefore more than enough time to select one of the actions without imposing pressure on the participants.

We added the following text to the Methods section “The right column contains two buttons and the time remaining to make a choice (i.e. 30 seconds). It is important to note that when the time runs out nothing happens, except that the participant is made aware that she should quickly make her choice (i.e. they received a warning on the screen). Participants were clearly informed about this before the start of the session. We only asked them to be prompt as delays in choosing would hinder the progress. ”

- Moreover, from the text it is not clear what was the time constraint. How many seconds did participants have to make their choice?

As explained in the last question, the time was 30 seconds. We added this information now in the Methods section (see added text above).

- How did you determine the end of the game? Random? Pre-determined?

The number of rounds was predetermined to be 50, however the participants were not informed about the exact number of rounds in order to avoid an end-of- game effect. We informed them nonetheless clearly that there will be multiple rounds and that the game will finish at an unknown moment yet within the time window of an hour. There was thus no deception. Note also that no counter of the rounds is shown on the display.

Fixing the number of rounds was done to ensure that each session has the same number of rounds, allowing for an easy averaging across the sessions. Varying the number of rounds for each session would introduce numerous complications and no real benefit.

Finally we do see from the results that the behavior does not change significantly at the very end, demonstrating that there was no end-of-game effect, as was anticipated.

- My understanding is that different treatments were conducted in different times. This is a violation of the random assignation procedure. Why do you think it is acceptable in this case? Moreover, why do you have 5 session with information but only four without? How was the sample size determined? Was it pre-determined?

This is a standard practice in laboratory experiments, as is exemplified by the many behavioral economics experiments in the literature. Such a setup is used for two reasons: i) organizational difficulties of working with large groups and ii) there is no reason to be concerned that the participant showing up at the different time slots are biased in any way. All the sessions were performed on Tuesday or Wednesday in two weeks of May, in the middle of the working days (from 10h to 16h).

Online experiments differ from laboratory experiments, yet recent publications have demonstrated that conclusions that can be drawn from them are equivalent (see e.g. Arechar, A. A., Gächter, S., & Molleman, L. (2018). Conducting interactive experiments online. *Experimental economics*, 21(1), 99-131.). Nonetheless, in online experiments it is easier to have a large number of players at the same time, yet there may be issues with drop-outs and time zones, as participants need to login at different moments in their daily schedules. Therefore, in the online experiment it is necessary to properly randomize the participants. However, this is not required for our laboratory experiments.

The main issue is not the sessions, but having sufficient participants for the statistical analysis: 64 would have been sufficient for both cases, but we were able to add a 5th one for TWI, obtaining 80 for that treatment. Time and organizational issues associated with the VUB facilities did not allow us to add an additional TWO session.

Results

- The sentence "Before the start of the actual game, the participants read through a tutorial and their understanding is tested with a small quiz. Concretely they are asked to answer how much they earn in a few situations, where each situation is defined by the actions taken by their neighbors and themselves." Should not be in the Results section, but in the methods.

We have moved these technical details to the Methods section. Thank you for the suggestion.

Discussion

- "The differences between both treatments might simply be a consequence of a stronger emotional response induced by the payoff difference and not a "rational" choice to act like the more successful neighbors": good point, especially because participants act under time pressure, and this favours emotional responses over deliberative ones.

As we explained before and clarified in the new version of the text, there is no real time pressure, so there is no reason to believe that emotional responses follow from that. The reason for adding this phrase is that based on the data we can say there is a correlation but that the underlying causes for the behavior cannot be derived. Our experiment does nonetheless show that the knowledge of the payoffs does change players behaviors and that this change in behavior is consistent with the assumption of imitation.

We thank you again for your comments and suggestions as they have allowed us to substantially improve the paper.

Reviewer: 2

Comments to the Author(s)

The manuscript reports an experimental study on the behaviour of people when making decisions in a Prisoner's Dilemma (PD) game played on a squared lattice network. The main research question is to measure whether participants take decisions imitating their most successful neighbour, i.e. the neighbour who earned the most in the previous round of the game. The authors implemented two experimental treatments, involving a total of 144 subjects, where the information about neighbours' previous payoff was either available or hidden, while neighbours' actions were always shown. Reported results depict that global cooperation levels are similar for both treatments while average earnings differ sensibly for cooperators and defectors. In the spirit of their research question, it appears that payoff differences influence the probability of changing action. Also, it is shown that probabilities to switch to another action are correlated to the number of cooperators in participants' neighbourhoods, in agreement with the previously observed behaviour of moody conditional cooperators.

The research topic is of interest and widely debated in the related literature, which is duly reported. However, the presentation, the statistical analysis, the presented results and the structure of the paper are largely below the threshold for considering the work as publishable. All text seems to be in a draft version: many grammar mistakes, typos, unfinished/unclear or extremely long sentences. The statistical analysis is not properly carried out: it looks like the authors used all observations, i.e. individual actions, as independent ones, while they actually have only 5 vs. 4 independent observations, i.e. the number of independent groups. Some results are meaningless for the main research question, which only arrives at Figure 5, and many more specific ones should be done to better, or properly, answer it. Finally, the structure should be strongly revised: KS test is unnecessarily explained, Figure 3 explanation comes before the one of Fig. 2d, Figure 6 and Table 4 are only mentioned in the discussion, the example variable explanation should not be in the results section, specific results in figures and tables should not be commented in the discussion. For these reasons, although I found the experimental procedure properly done and robust and the leading research question adequate, the manuscript cannot be considered for publication in its present version and a future re-submission is suggested.

We apologize for the errors and inconsistencies. The paper has been fully revised so that it meets the required standards. Your comments have allowed us to substantially improve the quality of the paper. Please find our answers to your specific comments below.

Here there are some more specific comments to improve the quality of the work:

- A deep revision of the text is required, I cannot list all the errors. Pages 3-4 of the Introduction and Methods section are particularly problematic.

We apologize for this. The paper has been fully revised and the additions are marked in red. Unnecessary parts, errors and repetitions were removed. The manuscript was also screened for remaining grammatical errors by different colleagues.

- Treatment names are introduced only in section 3 while they are already mentioned in the previous section.

Thank you for pointing this out. We have now introduced the names of the two treatments at the end of the introduction.

- It is either prisoner's dilemma or prisoners' dilemma.

Thank you for pointing this out. We corrected it in the new version and introduced the prisoner's dilemma and its abbreviation PD once. Afterwards PD is consistently used.

- The definition of Q^* in the Introduction is too technical and it should be moved to the methods/results, if it is really needed. However, it does not add much to the specific research question. It is not specified which is the used topology there.

We have now move this Q^* explanation to the Methods section as requested. This improves indeed the readability of the manuscript. We mentioned explicitly the topology at the beginning of the Methods section : a 4x4 square lattice with periodic boundary conditions.

- The KS test may be not enough or adequate for a complete statistical analysis. I would recommend to use the Mann-Whitney test too/instead. However, as I mentioned before, the number of independent observations is 9 and the statistical analysis on cooperation levels and earnings, if really needed, should be done with them, considering average group levels during the entire experiment. Having p-values of the order of $10^{(-133)}$ with 144 subjects is simply impossible. All tables and GLM have the same issue.

The KS test is necessary since the objective is to compare more (also spread and shape) than just the medians, which is what a Mann-Whitney test does. When comparing the medians there is again no significant difference in the distribution of cooperation ($p=0.1377$), however it is significant for the distribution of earnings (0.0102). As there are only a very few ties, the KS-test provided the best way to assess whether the distributions are significantly different.

For our analysis into the motivation for cooperation (imitation or not) we consider each decision made by a participant as a data point (thus 144×50 data points). We want to see if there is a correlation between having or not having payoff information and a behavioral change. To answer this question each participant's action counts as an independent observation. As a consequence, the observed p-values are perfectly possible. Clearly, as we also show in Table 4, decisions are based on whatever happened in a previous round, and we are fully aware of it since we show in Figure 5 and 6 what happens after a C or D action in the previous round.

- Figure 4 results are unclear. It is not clear which observation was discarded, and how. Again, instead than SD, error bars should be plotted, considering group averages as independent observations and normalising according to them, i.e. dividing by 4 or 5, if needed.

We thank the reviewer for this suggestion and we have now put the graph with error bars (standard error of the mean) instead.

- The example variable is a good sanity check that can be moved to the supplementary material and just mentioned in the main manuscript. The same applies to the random variable, if needed.

We have moved the discussion of the random, example variable and the GLM model to the Methods section.

- in the Appendix, what do you mean with "balanced participants"? I guess they were gender-balanced, or is it another kind of balance? In any case, more specific statistics on participants by treatment should be included in a dedicated section of the supplementary material, as well as the days of the experiments.

Yes, we meant gender balanced. This has now been clarified and we added the detailed statistics about the participants in the Appendix.

- references have to be revised, such as 27 or 29. There is not a standardised style and all titles are missing, making their search quite difficult.

We have adopted the Vancouver reference style, which appears to be the policy of the journal. The titles are thus now included.

- finally, I don't understand why the authors focused their study on average values instead than on individual behaviours; especially in an experimental study where the number of subjects is very small. An analysis on specific behavioural types of participants could have been much more intriguing and adequate than looking at, sometimes tiny, average differences between treatments. It could be the case that the behaviour of imitating the most successful neighbour exists, even to a larger extent, only for a certain percentage of subjects, as well as for the moody conditional behaviour.

We thank the referee for this comment. The main goal of this manuscript was to see whether imitation occurs with the spatial PD, a question, as you as well point out, that remains unanswered. For this we needed to see results over many players, which explains the focus on average values. In a follow-up work we are considering the individual behavior of each participant in a novel manner. This result was not added to not confound messages.

We thank you again for your comments and suggestions as they have allowed us to substantially improve the paper.

Appendix B

Response to reviewers

Thank you for accepting our work. Please find attached our responses to your final comments.

Reviewer: 1

Comments to the Author(s)

I thank the authors for addressing my comments. I think that the paper has been definitely improved. At the same time, I am not convinced by the way my comment regarding the risk of inducing time pressure has been addressed. The authors' response is that inducing people to respond within 30 seconds does not induce time pressure because other experiments used 30 seconds to induce time delay. This logic does not work because it fundamentally misses the point of time pressure vs time delay techniques. The number of seconds is not really the point of these techniques. The point is the feeling that participants sense when they have to answer *within* a certain amount of time vs *after* a certain amount of time. In one case they feel under pressure, in the other case they feel relaxed. Of course, the extent to which they feel under pressure may vary; so a time pressure of 10 seconds is *quantitatively* stronger than a time pressure of 30 seconds. However, also 30 seconds is *qualitatively* a time pressure. Therefore, I think that the authors have to re-address this point in a better way.

(En passant, let me note that it remains unclear to me why the authors introduced a time pressure in their experiment. To me, there was no need. But perhaps I'm misunderstanding something.)

We understand this reasoning but believe that any qualitative pressure felt in our experiment does not influence the observations: If any anxiety exists concerning the amount of time at the beginning, this disappears over the rounds as participants progress through the experiment. Note that they are informed that they play an undetermined number of rounds for an hour. The reason for explicitly mentioning the time on the screen is to ensure that rounds are not stalled by one of the participants and the pace is kept. Moreover, they were informed that nothing would happen when the time reached 0, giving them another reason not to feel pressured.

Giving them 30 seconds was actually shown to be more than sufficient time in another work published by one of the authors (see ref [1]). We see there that the distribution has an exponential tail with a median close to 3 seconds. Therefore although 10 seconds would impose pressure the distribution of reaction times drops so fast that there is no reason to believe that this pressure would still be there at the limit of 30 seconds.

As you can see in Figure 3 in the supporting information, the reaction times distribution of participants in our experiment also has an exponential tail with a median close to 3 seconds. Only a very small fraction of participants needs more than 30 seconds making this a good threshold for responding, while ensuring that the tempo of the experiment is maintained and people do not get bored or distracted while participating.

We added more explanation about the decision time to the Supplementary Material to avoid further confusions. In addition we expanded the reasoning for introducing the timer in the methods section. We hope this now satisfies your concern.

[1] Gallotti, Riccardo, and Jelena Grujić. "A quantitative description of the transition between intuitive altruism and rational deliberation in iterated Prisoner's Dilemma experiments." *Scientific reports* 9, no. 17046 (2019).

Reviewer: 2

Comments to the Author(s)

The manuscript, with respect to its previous version, largely improved from all points of view. All the introductory part is now much clearer and results, as well as their discussion, have been polished. Also, the authors satisfactorily addressed most of my previous comments. I am still a bit skeptical about the use of all observations as independent ones but I guess it is the best they can do. The paper still provides important evidences, perhaps not statistically significant as they state, to the research topic and it can be now considered for publication after addressing the few minor remarks reported below.

Thank you for accepting our work. Please find the answers to your comments below.

- Since the authors decided to not implement Wilcoxon test, it is unnecessary to talk about it in the Methods section.

We have removed this sentence from the methods section.

- The axes in Figures 3 should be the same to encourage the reader to a proper comparison. At least, when considering pairs of row panels.

We have now adjusted the axes of the bottom panel in Figure 3 so that comparison is made easier.

- The authors should state what they intend as independent observation, i.e. individual action, in the Methods section and I suggest them to get rid of those fishy p-values just stating something like $p < .0001$ instead.

We have updated the p-values and mentioned in methods that for Figure 3 and Table 1 each action of a participant is considered to be a data point.

- The authors could mention in the conclusions more insightful remarks for a follow-up research on the related topic as the framework of behavioral types or, generally speaking, a more detailed analysis that can be done at the individual level. What they found looking at the

"average" behavior can be largely heterogeneous at the individual one; as it is already visible in their work but it is not part of the main research question.

We fully agree, as before, that this is an interesting avenue. As mentioned in the previous response, we are working on a new manuscript in relation to this topic. Since you insist, we also added a small paragraph about this issue in the discussion section of this paper.